# Complementary and Alternative Medicines Used by Middle-Aged to Older Taiwanese Adults to Cope with Stress during the COVID-19 Pandemic: A Cross-Sectional Survey

**DOI:** 10.3390/healthcare10112250

**Published:** 2022-11-10

**Authors:** Yo-Yu Liu, Yueh-Chiao Yeh

**Affiliations:** 1Master’s Program in Natural Healing Sciences, Department of Natural Biotechnology, Nanhua University, Chiayi 622, Taiwan; 2Doctoral Program in Management Science, Department of Business Administration, Nanhua University, Chiayi 622, Taiwan

**Keywords:** COVID-19 pandemic, middle-aged adult, older adults, complementary and alternative medicines, music therapies, religion, vegetarian diets

## Abstract

Background: This study aimed to investigate the factors influencing the use of complementary and alternative medicines (CAMs) to manage stress during the COVID-19 pandemic in Taiwan. Methods: A cross-sectional survey was administered to community-dwelling adults between the ages of 46 and 75 years, and a total of 351 participants completed the questionnaire. Log-binominal regression analyses were fitted to explore the factors associated with the use of CAMs. Results: The mean age of the participants was 57.0 years, and 67.0% reported that they had used CAMs within the past three months. Middle-aged adults were more likely to use CAMs than late middle-aged adults and older adults (*p* < 0.001). Overall, the major CAMs utilized to relieve psychological stress were music therapies (37.6%), massage (31.1%), spinal manipulation (25.1%), relaxing therapies (24.2%), and reading scriptures or *The Bible* (23.9%). Religion and vegetarian diets were the most important factors influencing participants to use CAMs, especially music therapies, massage, and reading scriptures/*The Bible*. Conclusions: CAM use was very prevalent among middle-aged adults in Taiwan; in particular, music therapies were the most favored activities for reducing stress. Population-specific mental health interventions using music can be developed to improve stress management outcomes during public health emergencies.

## 1. Introduction

According to the World Health Organization (WHO), Taiwan has been an aging society since 2018, with people aged 65 years or above accounting for 14% of Taiwan’s total population, and the country will enter the super-aged society era by 2025 (more than 20% of the population above the age of 65) [1,2]. As is well known, the health of middle-aged to older adults (the population aged 45 and above) is the most important concern, which will require increasing attention in the future [2,3,4,5,6], as the incidence of chronic disease rises dramatically with age and the majority of patients with a chronic disease are over the age of 65 years. Studies have also reported that the increased prevalence of chronic disease in middle-aged to older populations has led to increased healthcare expenditure, as well as having impacts on their physiological health and psychological stress, eventually affecting their quality of life [7,8,9].

The recent COVID-19 pandemic, caused by the severe acute respiratory syndrome coronavirus 2 (SARS-CoV-2), has not only caused over 6 million deaths globally, but has also triggered a wide range of psychological issues, such as anxiety, depression, and acute stress symptoms in healthcare workers as well as those who under self-quarantine or self-isolation [10,11,12]. Accordingly, adults aged 45 years and older with chronic diseases or illness were more worried about the spread of infection and were more likely to experience loneliness and negative mental health consequences during the COVID-19 pandemic period, particularly in Asia [13,14]. A growing number of studies have indicated that people worried about becoming infected with COVID-19 and those who experienced considerable levels of stress considered the use of complementary and alternative medicines (CAMs) to enhance the balance of their body, mind, and spirit during the first wave of the COVID-19 pandemic [15,16,17]. CAMs are defined as diagnostic procedures, self-care management, and treatment practices, being non-medical interventions that are not generally considered part of conventional medicine. They can generally be classified into several types, such as nutritional, psychological, physical, and combination approaches [4]. Studies have indicated that up to 80% of the general population used at least one CAM in the past year. The reasons why they seek CAMs included enhancing disease-fighting properties through antioxidant supplements, improving disease-specific quality of life, increasing vital energy flow, and improving their physical and mental health [18,19,20]. Traditional Chinses medicine (TCM) treatments or herbal products, muscle relaxation techniques (e.g., yoga, massage, tai chi, dance, and exercise), functional food and dietary supplements (e.g., probiotics and vitamin D), mindfulness activities (e.g., Ayurveda and Reiki), and aromatherapy were the most popular CAM interventions used at home to ease the symptoms induced by COVID-19 infection, support the immune system to avoid infection, or promote psychological wellbeing in order to return to normal life during this stressful time [10,20,21,22,23].

Non-invasive CAM modalities have been found to have a beneficial impact on altering stress-related physical and psychosocial outcomes, or improving mental wellbeing in different populations [3,9,18,24,25]. Accordingly, certain factors have been reported to be associated with the CAM utilization, such as a strengthened immune system to alleviate infectious diseases, altered nervous system activity to improve mental wellbeing and quality, and regulation of the hypothalamic–pituitary–adrenal (HPA) axis to relieve psychological stress [22,26,27]. Despite the above, most studies have focused on CAM utilization in adult or pediatric populations with no particular focus on CAM use among the community-dwelling middle-aged to older adult population. Taken together, research on the prevalence of usage of different CAMs for emotional management during the COVID-19 public health emergency is sparse in Taiwan. Moreover, the potential factors in relation to CAM utilization also remain under investigation. Therefore, the main objective of this study was to compare the prevalence of CAM utilization among three age groups—middle-aged adults (ages 46–55), late middle-aged adults (ages 56–65), and older adults (ages 66–75)—in order to examine the influencing factors associated with the most-used CAMs for self-care management and coping with psychological distress during the global pandemic period in Taiwan. The outcomes of this study are important, in terms of allowing healthcare providers and decision makers to evaluate healthcare practices and develop strategies that improve health outcomes in middle-aged to older adults.

## 2. Materials and Methods

### 2.1. Study Design and Participants

A cross-sectional survey was conducted in participants from May 2020 to February 2021, when the COVID-19 pandemic had not had a serious outbreak in Taiwan, as the number of confirmed cases was not too high, compared with other countries. Adults aged between 46 and 75 years from different regions of Taiwan were invited to participate. Participants were excluded if they were under 46 years old, unable to complete an interview, or had any of the following disorders: severe mental disorders, including high levels of depression or anxiety symptoms; taking antidepressants; or at acute risk for suicide and needing immediate care. All surveyed participants were informed of the study protocol and were included in the study after providing written informed consent. Ethical approval was granted by the Human Research Ethics Committee of National Chung Cheng University, Taiwan (Approval number: CCUREC109043001 2020–2021) before the start and was in accordance with the Declaration of Helsinki.

### 2.2. Cross-Sectional Survey Sampling

Eligible participants recruited from three city zones (north, central, and south) in Taiwan were sampled by randomly selecting ten communities per zone for a total of thirty communities in this survey. Sample size calculations for this individual cross-sectional studies were determined by the Sample Size Calculator [28], a public service of Creative Research System survey software, according to the population between years 46 and 75 in Taiwan. To obtain accurate findings, the suitable sample size for interviewing is 384 to allow for a sampling error of 5% (with 95% confidential level) to reflect the demographic profile of Taiwan. In summary, 450 people were recruited and 351 participants completed a valid response. The total response rate was 78% in this study.

### 2.3. Development of the Survey Form

The questionnaire was developed in a 3-step procedure of preliminary drafting according to our previous published paper [20], revision based on five expert opinions, and final editing. Questionnaires on CAM uses within the past three months was classified into six categories according to the classification system for various modalities from the National Center for Complementary and Integrative Health (NCCIH) [4]. They were namely nutritional approaches (e.g., herbs, probiotics, dietary supplements, special diets, or microbial-based therapies), psychological approaches (e.g., meditation, music therapies, or relaxing therapies), physical approaches (e.g., acupuncture, massage, or spinal manipulation), combinations of psychological and physical or psychological and nutritional approaches (e.g., yoga, tai chi, qigong, art therapies, aromatherapy, dance therapies, or mindful eating), other complementary health approaches (traditional Chinese medicine (TCM), homeopathy, naturopathy, Ayurvedic medicine, or functional medicine), and other (e.g., far infrared rays, reading scriptures or *The Bible*, or fortune-telling). The reliability and validity of the questionnaire were 0.863 and 0.90, respectively.

### 2.4. Survey Distribution

The questionnaire was completed, on average, within 20 min. One researcher gave an explanation of the survey method to each participant prior to filling out the questionnaire to minimize confusion. Another researcher checked all questionnaires for completeness and errors at the end of survey. All participants were informed of the objectives, questionnaire development procedure, and survey completion method, and that personal information would be protected and that use of questionnaire results would be limited to academic means.

### 2.5. Data Setting

Each participant was asked to complete a paper-based questionnaire consisting of questions on their demographic characteristics (gender, body mass index, marital status, number of children, education level, occupation, and religion), lifestyle characteristics, perceived health status and health conditions, and previous use of CAMs to manage stress during the COVID-19 pandemic period (see Appendix A. A file for the final version of the questionnaire).

### 2.6. Statistical Analysis

Descriptive statistics, including means and standard deviations (SD) for continuous data, and counts and percentages for categorical variables were calculated as appropriate. Participants were further classified into three age groups: Middle-aged adults (age 46–55, *N* = 177), late middle-aged adults (age 56–65, *N* = 108), and older adults (age 66–75, *N* = 66). Comparisons between categorical variables were accomplished using Pearson’s Chi-square test or one-way analysis of variance (ANOVA). Log-binomial regression analyses were conducted in order to calculate the adjusted prevalence ratios and 95% confidence intervals for different variables and to evaluate the independent factors associated with the use of CAM modalities. All analyses were conducted using the PASW Statistics software for Windows (Version 18.0; SPSS Inc., Chicago, IL, USA). A probability value of less than 0.05 was considered to denote statistical significance.

## 3. Results

Table 1 provides the demographic statistics of the 351 included respondents (78% completion rate). Of the participants, the mean age was 57.0 years and females were the majority (53.8%). Among the three age groups, most of the older adults had more than two children (72.7%, *p* < 0.001), had attained the education level of high school degree or below (50.0%, *p* < 0.001), and were retired or did not work (74.3%, *p* < 0.001). However, more middle-aged adults had a college degree or above (72.3%, *p* < 0.001), were in a skilled/professional occupation (29.3%, *p* < 0.001), exercised irregularly (54.2%, *p* = 0.011), drank functional beverages irregularly (21.1%, *p* = 0.034), and chose vegetarian diets irregularly (71.8%, *p* < 0.001). In Table 2, the results show that more than half of the participants perceived their health status to be poor/very poor (54.4%) and had a history of chronic disease (59.5%). Of the participants, many older adults had chronic disease (75.8%, *p* = 0.007), especially hypertension (37.9%, *p* < 0.001), hyperlipidemia (24.2%, *p* < 0.001), and sleep disorders (19.7%, *p* < 0.001). Regarding the use of prescription medication, 72.7% of the older adults took more than one prescription medicine, which was statistically higher when compared with the other two groups of participants (*p* = 0.046). The use of hypolipidemic agents (21.2%, *p* < 0.001), anti-hypertensives (17.6%, *p* < 0.001), and hypnotics (12.1%, *p* = 0.048) were also significantly higher in older adults.

During the three months before the interview, the overall prevalence of the use of CAMs to cope with stress during the COVID-19 pandemic was 67.0%. Notably, more middle-aged adults (72.9%) reported that they had used CAMs to relieve their psychological anxiety, compared to the other two groups of participants (*p* = 0.045). The most common CAM types were music therapies (37.6%), massage (31.1%), spinal manipulation (25.1%), relaxing therapies (24.2%), and reading scriptures/*The Bible* (23.9%).

Among the different types of CAMs, middle-aged adults significantly preferred to use music therapies (e.g., music listening) (48.0%, *p* < 0.001), massage (38.4%, *p* = 0.008), art therapies (e.g., coloring activities) (24.9%, *p* = 0.004), meditation (24.9%, *p* = 0.010), reading scriptures/*The Bible* (23.7%, *p* = 0.040), naturopathy (23.7%, *p* = 0.010), yoga (19.2%, *p* < 0.001), far infrared rays (14.1%, *p* = 0.014), qigong (8.5%, *p* = 0.046), fortune-telling (11.9%, *p* = 0.014), and functional medicine (8.5%, *p* = 0.046) when compared to late middle-aged adults and older adults (Table 3).

Table 4 displays the log-binomial regression analyses for CAM use and each of the five top CAMs used to help support the mental stress of participants during the COVID-19 pandemic period, with respect to their independent variables. First of all, religion (Christian/Other, PR = 1.43; 95% CI: 1.11–1.84, *p* = 0.006), vegetarian diets (irregularly, PR = 1.24; 95% CI: 1.01–1.52, *p* = 0.038), prescription medication use (PR = 1.26; 95% CI: 1.08–1.47, *p* = 0.003), and allergic rhinitis (PR = 1.27; 95% CI: 1.03–1.56, *p* = 0.027) were four positive factors associated with the use of CAMs, whereas late middle-aged adults (PR = 0.85; 95% CI: 0.72–1.00, *p* = 0.049) and hyperlipidemia (PR = 0.62; 95% CI: 0.42–0.90, *p* = 0.012) were two negative factors.

Regarding the associated factors influencing the participants to use the top five CAMs, religion (Buddhism, PR = 1.86; 95% CI: 1.27–2.72, *p* = 0.001 and Christian/Other, PR = 2.16; 95% CI: 1.28–3.66, *p* = 0.004) and vegetarian diets (irregularly, PR = 1.56; 95% CI: 1.05–2.30, *p* = 0.027) were positive factors affecting with the use of music therapies, whereas age (Late middle-aged adults, PR = 0.58; 95% CI: 0.42–0.80, *p* = 0.001) and hyperlipidemia (PR = 0.42; 95% CI: 0.19–0.95, *p* = 0.036) were two negative factors. It is worth noting that those who were middle-aged adults had specific religion, chose vegetarian diets (irregularly, PR = 2.31; 95% CI: 1.36–3.92, *p* = 0.002 and regularly, PR = 2.67; 95% CI: 1.30–5.43, *p* = 0.007), and smoked cigarettes irregularly (PR = 1.67; 95% CI: 1.09–2.53, *p* = 0.019) were more likely to use massage.

Remarkably, those who did not have a particular type of occupation (PR = 1.79; 95% CI: 1.06–3.03, *p* = 0.029) and drank alcohol regularly (PR = 1.98; 95% CI: 1.02–3.84, *p* = 0.045) tended to use spinal manipulation. Conversely, those who perceived good/excellent health status (PR = 0.59; 95% CI: 0.35–0.99, *p* = 0.044) were less likely to use this CAM. Buddhists (OR = 2.90, 95% CI: 1.50–5.62, *p* = 0.002) and Taoists (OR = 2.86, 95% CI: 1.43–5.73, *p* = 0.003) were more likely to use relaxing therapies to have less stress. Finally, Buddhists (OR = 7.62, 95% CI: 2.84–20.48, *p* < 0.001), Christian/other (OR = 8.47, 95% CI: 2.90–24.7, *p* < 0.001), and chose vegetarian diets regularly (OR = 2.35, 95% CI: 1.23–4.49, *p* = 0.010) were positive factors associated with reading scriptures/*The Bible*. Interestingly, those who were perceived as poor/very poor (OR = 0.55, 95% CI: 0.36–0.84, *p* = 0.005) and with a good/excellent (OR = 0.52, 95% CI: 0.33–0.82, *p* = 0.005) health status were less likely to read scriptures/*The Bible* compared with those reporting a fair perceived health status.

## 4. Discussion

To date, there have been very few studies regarding the demographic and clinical influencing factors associated with the use of complementary and alternative medicines as stress relief strategies during the first lockdown of the COVID-19 pandemic, considering healthy middle-aged (age 46–55 years), late middle-aged (age 56–65 years), and older adults (age 66–75 years) in the community as the research population in Taiwan. In this study, we aimed to examine the prevalence of CAM utilization from mid-2020 to early 2021, in order to shed light on the impact of stress during the COVID-19 pandemic among middle-aged to older adults. Overall, 67.0% of the study population reported that they had used at least one CAM modality during the past three months. Middle-aged adults tended to use different kinds of CAMs (72.9%) to manage stress more than late middle-aged (63.0%) and older adults (57.6%). Interestingly, music therapies was the most popular CAM to improve mental health. These findings may help further understanding regarding the use of CAMs for the protection of mental health and to provide information for positive psychological prevention and intervention strategies in the context of the COVID-19 global pandemic.

During the past two years, the pandemic restrictions have had negative impacts on many people. Most people have experienced the fear of viral infection and increased anxiety or depression symptoms induced by social isolation due to lockdowns, loneliness, economic disruptions, and the adoption of new ways of working [17,29]. Epidemiological studies have indicated that the middle-aged to older population were more affected by the COVID-19 pandemic than young adults and children. Particularly, older adults are generally more susceptible to severe illnesses and have a high mortality rate due to various chronic diseases [3,11]. In the past few decades, Taiwan has been faced with challenges associated with the issue of an aging population. The majority of patients with a chronic ailment are over the age of 65 years, and more than 50% of the older population had more than two chronic diseases [9,30]. Our data showed that the chronic diseases with higher prevalence among older adults were hypertension (more than 30%) and hyperlipidemia (more than 20%), consistent with other extensive studies conducted over the past several years in both Taiwan and other countries [3,24,31,32]. Notably, work pressure or other stress-induced high blood pressure was related to mental health problems and disease-specific quality of life. Other chronic diseases also have been linked to diet; in terms of dietary modification, increased fruit and vegetable intake can decrease the risk of hypertension, obesity, and type 2 diabetes [9,33]. Similar to previous reports, our data indicated that middle-aged adults significantly preferred to choose vegetarian diets (79.1%), use CAMs (72.9%), and had less chronic disease (1.2 ± 1.4) than the older adult population during the COVID-19 pandemic. These results can be used in further studies in order to investigate whether healthy eating habits can enhance the health condition or not, especially under pressure. Furthermore, the question regarding whether a vegetarian diet is a predictor for the prevention of disease and maintenance of well-being remains under exploration.

CAM usage has been reported to be significantly correlated with gender, age, education level, health status, and income [3,15,16]. Recent studies have suggested that middle-aged to older adults tended to seek different kinds of CAM to help them improve the harmony between their mind, body, and spirit [6,18,24]. Middle-aged adults, especially women and those with a higher education level, tend to search for more valuable sources of information to use different kind to CAMs and share their experiences with their friends [20,30,34]. In Taiwan, the prevalence rate for CAM use in the previous 12 months was 38 to 86% among people aged 18 years and above to maintain their health. The most commonly used CAMs were dietary supplements, massage, spinal manipulation, and traditional Chinese medicine (TCM) [35,36]. Our data showed that the prevalence rate of CAM use to cope with stress in the older adults (57.6%) was significantly lower than in the middle-aged adults (72.9%) during the COVID-19 pandemic. Regarding the reason why older people had lower CAM utilization, one reason may be the risk of virus infection for older adults influence them to go out during the COVID-19 outbreak or visit CAM institution. Another possible reasons might be a lack of trained CAM practitioners to give them accurate advice, such as how to use aromatherapy, physical therapy, art therapies, and so on [19,35]. The other reason might be influenced by healthcare spending regarding the use of CAMs. Finally, older adults generally have poorer health which makes them or their healthcare givers more likely to consider CAM use for the treatment of their diseases rather than for coping with mental stress during the COVID-19 pandemic. However, further investigations can be performed in order to understand the causal relationships related to this question.

Additionally, it is difficult to understand the correlation of CAM uses with the COVID-19 pandemic in Taiwan, especially since most of the surveys had different definitions of CAM. For example, the application of various TCM treatments to treat COVID-19 pandemic-induced adverse psychological effects has also been shown to be effective [16,17,35]. Correspondingly, a lower education level and older age might reflect low socio-economic status, thus influencing individuals to use other CAMs. Furthermore, most research only focused on some specific CAM types used in their interested target population; therefore, further investigation is required to explore the correlation of prevalence and patterns of CAM use with the reasons they are used in Taiwan. Another possible reason that might cause our result to have a bias is the sample size of the older adult group. According to the statistics of Taiwan’s population, the ratio of our three study groups should be 40:38:22; however, our population ratio is 50:31:19. One possibility is the lower response rates for the 60+ age group misrepresents the actual results of Taiwanese older adults’ choices of CAM uses. However, it is difficult to distinguish whether these results can be interpreted in relation to the COVID-19 pandemic or reflect the Taiwanese population’s preference for CAM treatments. Notably, in our questionnaire, we have specifically asked our respondents which CAM they used to cope with stress during the COID-19 pandemic period. Therefore, our findings should be interpreted in relation to the COVID-19 pandemic rather than a reflection of the Taiwanese population’s preference for CAM treatments. Further investigations should be performed in order to explore the causal relationships related to this issue.

Listening to music is a simple stress relief means that has been widely accepted by the public [22,27]. In this study, middle-aged adults preferred to use music therapies (e.g., listen music) to relieve their stress during the COVID-19 pandemic, especially those who had specific religions (e.g., Buddhism and Christian/other) and chose vegetarian diets. One recent study found that mindfulness-based music listening can regulate negative emotions induced by the COVID-19 pandemic in young adults [37]. Whether the effect of listening to religious music is similar to mindfulness-based music or mediation music on relieving a body’s physiological stress is worth investigating. As is well known, individuals remain in the workforce and have the highest psychological stress responses related to family, economic, and health issues in this stage of life [8,34]. It has also been found that music listening interventions or online music therapies have effects on reducing anxiety, enhancing mood, improving mental health, and supporting subjective wellbeing, not only for coping with diagnosed diseases but also for combating COVID-19-induced mood disturbance or mental stress [21,26,38]. Some types of music may actually increase negative emotions, although this differential effect of music has not yet been adequately tested. Liu and colleagues’ study indicated that sad music increases the young university students’ negative mood states, whereas calm and happy music can effectively adjust participants’ adverse emotions [37]. Interestingly, another report revealed that differences in age and musical expertise will affect individuals’ emotion recognition in music [39]. Further studies can be designed to evaluate the short- or long-term intervention effects of different types of musical activities and the duration of music listening practices on the causal mechanism of physiological and psychological responses for ameliorating stress, reducing negative emotions, and moderating the quality of life in middle-aged adults.

Our data indicated that those adults who were middle-aged and had a specific religion smoke cigarette irregularly, chose vegetarian diets, and preferred to use massage for coping with stress, whereas those who drank alcohol regularly and did not have a particular type of occupation preferred to use spinal manipulation. Previous studies have demonstrated that massage therapy or aromatherapy massage can alleviate stress-induced psychophysiological discomfort, lower blood pressure and heart rates, decrease stress hormone levels, reduce fatigue, and improve sleep quality [6,31]. A recently published article has indicated that chiropractic care or spinal manipulation can provide management for musculoskeletal disorders; however, the inappropriate information from social media and the internet provided by chiropractic regulatory agencies which claimed that it can enhance immunity to protect against COVID-19 should be strictly noted by healthcare providers [40]. One cross-sectional survey also indicated that massage, TCM, and chiropractic treatments can cause adverse side-effects [34]. Another systematic review and meta-analysis results demonstrated that TCM and acupuncture has been frequently used to successfully treat patients’ symptoms caused by COVID-19 [41]. Our data showed that they were not the main options in middle-aged and older adults during the early stage of COVID-19 in Taiwan. Since the infection rates were significantly augmented after mid-May 2021, further studies can investigate whether TCM or acupuncture was used in patients when they were infected by COVID-19. Additionally, another study revealed that mindfulness practices had positive effects on emotional wellbeing and general health in the context of the COVID-19 pandemic [30]. Another publication also emphasized that mindfulness-based music listening can improve an individual’s negative emotions relating to COVID-19 [37]. Whether those who used these CAM practices had special religions, occupation, lifestyle, or eating habits which affected their choices remain unclear. Due to the observational nature of this study, further evidence-based systematic studies may be performed to explore the causal relationships related to this issue.

This study had certain limitations that should be taken into consideration when interpreting these results. First, due to the cross-sectional nature of the study, causality could not be assessed. Further evidence-based systematic studies or intervention research should be performed to explore the possible effects in middle-aged and older adults. Second, the lack of more details about the effectiveness of CAM usage limited our ability to determine the most common possible means to relieve stress among middle-aged adults during the COVID-19 pandemic. Further qualitative interviews can be performed to investigate the participants’ responses. Third, as the data were collected in 2021, the pattern of CAM utilization may have changed, especially as specific treatments were available and the severity of illness in people who got the virus was much lower at that point. Additionally, our small-scale survey study has provided helpful outcomes for developing strategies to cope with stress during the COVID-19 pandemic; however, further large-scale cross-sectional and longitudinal studies can be designed to understand the changes of CAM uses after the decline in COVID-19 infections or the increase of vaccination rates. Fourth, the population aged 75 years or older tend to have more chronic diseases and the problems related to polypharmacy may affect their choices to use CAMs [42]. Therefore, this study only focused on the middle-aged and older adults ages 46 to 75. This may lead to biased research results. Finally, with the current data, we could not evaluate the stress-reducing effects of CAM use, limiting our ability to understand the possible associated psychological mechanisms. Further studies could classify participants into high- or low-risk of stress groups to compare their CAM uses during the period of the COVID-19 pandemic.

## 5. Conclusions

Based on this cross-sectional survey, it shows that age, occupation, religion, and vegetarian diets are important factors that influence CAM usage in middle-aged adults and older adults in order to cope with their mental stress during the COVID-19 lockdowns. Taken together, the most-used CAMs were music therapies, massage, and spinal manipulation. We also discovered that CAM usage was popular with middle-aged adults in order to cope with stress during the outbreak of COVID-19. Preventive is better than curative treatment; therefore, future investigations on the safety and benefits of such interventions are important to minimize the adverse effects and ensure the efficacy of CAMs. Futhermore providing education to healthcare professionals when designing policies in the context of public health crises is important as well. Developing intervention strategies using music therapies to regulate emotional stress related to global disease outbreak is helpful for the entire population to cope with COVID-19-induced psychological stress.

## Figures and Tables

**Table 1 healthcare-10-02250-t001:** Basic demographic statistics of the study participants (*N* = 351).

Variable	*N* (%)				*p*-Value
	Total Population 351 (100)	Middle-Aged Adults 177 (50.4)	Late Middle-Aged Adults 108 (30.8)	Older Adults 66 (18.8)	
**Age (years), mean (SD)**	57.0 ± 7.6	50.9 ± 2.8	59.4 ± 2.8	69.5 ± 2.7	<0.001 **
**Gender**					0.433
Female	189 (53.8)	101 (57.1)	56 (51.9)	32 (48.5)	
Male	162 (46.2)	76 (42.9)	52 (48.1)	34 (51.5)	
**BMI (kg/m^2^), mean (SD)**	23.2 ± 3.2	23.3 ± 3.2	23.3 ± 3.0	23.1 ± 3.5	0.927
**Body mass index (BMI)**					0.495
Normal weight	207 (59.0)	108 (61.0)	83 (58.3)	36 (54.5)	
Underweight	11 (3.1)	3 (1.7)	4 (3.7)	4 (6.1)	
Overweight	93 (26.5)	44 (24.9)	28 (25.9)	21 (31.8)	
Obese	40 (11.4)	22 (12.4)	13 (12.1)	5 (7.6)	
**Marital status**					0.003 **
Single	44 (12.5)	24 (13.8)	13 (12.0)	7 (10.6)	
Married	253 (72.1)	136 (76.7)	78 (72.3)	39 (59.1)	
Divorced/Widow/Other	54 (15.4)	17 (9.5)	17 (15.7)	20 (30.3)	
**Number of children**					<0.001 **
0	61 (17.4)	37 (20.9)	15 (13.9)	9 (13.7)	
1	80 (22.8)	46 (26.0)	27 (25.0)	7 (10.6)	
2	147 (41.8)	71 (40.1)	50 (46.3)	26 (39.4)	
≥3	61 (17.4)	23 (13.0)	16 (14.8)	22 (33.3)	
**Education level**					<0.001 **
Elementary school or below	9 (2.6)	4 (2.3)	0 (0)	5 (7.6)	
Middle school	23 (6.6)	5 (2.8)	4 (3.7)	14 (21.2)	
High school	91 (26.0)	40 (22.6)	37 (34.2)	14 (21.2)	
College/University	153 (43.7)	80 (45.2)	45 (41.7)	28 (42.4)	
Masters or above	74 (21.1)	48 (27.1)	21 (19.4)	5 (7.6)	
**Occupation**					<0.001 **
No/Retired	108 (30.8)	24 (13.6)	35 (32.4)	49 (74.3)	
Teacher/Public employee	69 (19.7)	48 (27.1)	18 (16.6)	3 (4.5)	
Salesmen/Attendance	42 (12.0)	29 (16.4)	10 (9.3)	3 (4.5)	
Skilled/Professional	85 (24.2)	52 (29.3)	27 (25.0)	6 (9.1)	
Other	47 (13.3)	24 (13.6)	18 (16.7)	5 (7.6)	
**Religion**					0.1
No	89 (25.3)	56 (31.6)	21 (19.4)	12 (18.2)	
Buddhism	148 (42.2)	74 (41.8)	48 (44.5)	26 (39.4)	
Taoism	89 (25.4)	36 (20.3)	30 (27.8)	23 (34.8)	
Christian/Other	25 (7.1)	11 (6.2)	9 (8.3)	5 (7.6)	
**Exercise**					0.011 *
No	96 (27.4)	55 (31.1)	29 (26.9)	12 (18.2)	
Irregularly	183 (52.1)	96 (54.2)	56 (51.8)	31 (47.0)	
Regularly	72 (20.5)	26 (14.7)	23 (21.3)	23 (34.8)	
**Smoking**					0.153
No	280 (79.8)	137 (77.4)	88 (81.5)	55 (83.4)	
Irregularly	37 (10.5)	24 (13.6)	11 (10.2)	2 (3.0)	
Regularly	34 (9.7)	16 (9.0)	9 (8.3)	9 (13.6)	
**Alcohol use**					0.204
No	251 (71.5)	119 (67.2)	83 (76.9)	49 (74.2)	
Irregularly	92 (26.2)	55 (31.1)	21 (19.4)	16 (24.3)	
Regularly	8 (2.3)	3 (1.7)	4 (3.7)	1 (1.5)	
**Betel nut chewing**					0.448
No	329 (93.7)	163 (92.1)	101 (93.5)	65 (98.5)	
Irregularly	20 (5.7)	13 (7.3)	6 (5.6)	1 (1.5)	
Regularly	2 (0.6)	1 (0.6)	1 (0.9)	0 (0.0)	
**Drink coffee**					0.098
No	113 (32.2)	56 (31.6)	35 (32.4)	22 (33.3)	
Irregularly	128 (36.5)	59 (33.4)	37 (34.3)	32 (48.5)	
Regularly	110 (31.3)	62 (35.0)	36 (33.3)	12 (18.2)	
**Drink tea**					0.292
No	64 (18.3)	38 (21.6)	13 (12.0)	13 (19.7)	
Irregularly	182 (52.0)	90 (51.1)	61 (56.5)	31 (47.0)	
Regularly	104 (29.7)	48 (27.3)	34 (31.5)	22 (33.3)	
**Drink functional beverage**					0.034 *
No	257 (73.2)	118 (73.2)	82 (75.9)	57 (86.4)	
Irregularly	74 (21.1)	74 (21.1)	20 (18.5)	8 (12.1)	
Regularly	20 (5.7)	20 (5.7)	6 (5.6)	1 (1.5)	
**Drink milk**					0.204
No	124 (35.0)	64 (36.2)	39 (36.4)	21 (31.8)	
Irregularly	189 (54.0)	101 (57.1)	53 (49.5)	35 (53.0)	
Regularly	37 (10.6)	12 (6.8)	15 (14.0)	10 (15.2)	
**Vegetarian diets**					<0.001 **
No	87 (24.8)	37 (20.9)	22 (20.4)	28 (42.4)	
Irregularly	239 (68.1)	127 (71.8)	74 (68.5)	38 (57.6)	
Regularly	25 (7.1)	13 (7.3)	12 (11.1)	0 (0.0)	

Note: Middle-aged adults = 46–55 years; late middle-aged adults = 56–65 years; older adults = 66–75 years. Value are frequencies (percentages), unless otherwise specified. SD: standard deviation. Data were analyzed using Pearson correlation or one-way analysis of variance (ANOVA). * *p* < 0.05; ** *p* < 0.01. BMI: body mass index is the weight in kilograms divided by the square of the height in meters. Normal: BMI 18.5–23.9 kg/m^2^; underweight: BMI ≤ 18.4 kg/m^2^; overweight: BMI 24.0–26.9 kg/m^2^; obese: BMI ≥ 27.0 kg/m^2^.

**Table 2 healthcare-10-02250-t002:** Health conditions and medication use in study participants (*N* = 351).

Variable	*N* (%)				*p*-Value
	Total Population 351 (100)	Middle-Aged Adults 177 (50.4)	Late Middle-Aged Adults 108 (30.8)	Older Adults 66 (18.8)	
**Perceived health status**					0.954
**Fair**	34 (9.7)	16 (9.0)	10 (9.3)	8 (12.1)	
**Poor/Very poor**	191 (54.4)	97 (54.8)	58 (53.7)	36 (54.5)	
**Good/Excellent**	126 (35.9)	64 (36.2)	40 (37.0)	22 (33.3)	
**History of chronic disease**					0.007 **
No	142 (40.5)	74 (41.8)	52 (48.1)	16 (11.3)	
Yes	209 (59.5)	103 (58.2)	56 (51.9)	50 (75.8)	
**Number of chronic diseases, mean (SD)**	1.2 ± 1.4	1.2 ± 1.6	0.9 ± 1.1	1.5 ± 1.4	0.039 *
**Hypertension**					<0.001 **
No	282 (80.3)	154 (87.0)	87 (80.5)	41 (62.1)	
Yes	69 (19.7)	23 (13.0)	21 (19.4)	25 (37.9)	
**Presbyopia**					0.389
No	283 (80.6)	139 (78.5)	87 (80.8)	57 (86.4)	
Yes	68 (19.4)	38 (21.5)	21 (19.4)	9 (13.6)	
**Myopia**					0.046 *
No	296 (84.3)	141 (79.7)	95 (88.0)	60 (90.9)	
Yes	55 (15.7)	36 (20.3)	13 (12.0)	6 (9.1)	
**Hyperlipidemia**					<0.001 **
No	317 (90.3)	166 (93.8)	101 (93.5)	50 (75.8)	
Yes	34 (9.7)	11 (6.2)	7 (6.5)	16 (24.2)	
**Sleep disorders**					<0.001 **
No	320 (91.2)	163 (92.1)	104 (96.3)	53 (80.3)	
Yes	31 (8.8)	14 (7.9)	4 (3.7)	13 (19.7)	
**Upset stomach**					0.594
No	324 (92.3)	164 (92.7)	101 (93.5)	59 (89.4)	
Yes	27 (7.7)	13 (7.3)	7 (6.5)	7 (10.6)	
**Allergic rhinitis**					0.393
No	328 (93.4)	163 (92.1)	101 (93.5)	64 (97.0)	
Yes	23 (6.6)	14 (7.9)	7 (6.5)	2 (3.0)	
**Diabetes**					0.19
No	329 (93.7)	166 (93.8)	104 (96.3)	59 (89.4)	
Yes	22 (6.3)	11 (6.2)	4 (3.7)	7 (10.6)	
**Prescription medication use**					0.046 *
No	142 (40.5)	78 (44.1)	46 (42.8)	18 (27.3)	
Yes	209 (59.5)	99 (55.9)	62 (57.4)	48 (72.7)	
**Number of prescription medication use, mean (SD)**	0.9 ± 1.4	0.8 ± 1.4	0.7 ± 1.1	1.4 ± 1.5	0.003 **
**Anti-hypertensives**					<0.001 **
No	287 (81.8)	156 (88.1)	89 (82.4)	89 (82.4)	
Yes	64 (18.2)	21 (11.9)	19 (17.6)	19 (17.6)	
**Painkillers**					0.916
No	306 (87.2)	153 (86.4)	95 (88.0)	58 (87.9)	
Yes	45 (12.8)	24 (13.6)	13 (12.0)	8 (12.1)	
**Upset stomach relief**					0.945
No	309 (88.0)	155 (87.6)	96 (88.9)	58 (87.9)	
Yes	42 (12.0)	22 (12.4)	12 (11.1)	8 (12.1)	
**Hypolipidemic agents**					<0.001 **
No	319 (91.1)	189 (96.0)	98 (90.7)	52 (78.8)	
Yes	31 (8.9)	7 (4.0)	10 (9.3)	14 (21.2)	
**Muscle relaxant**					0.442
No	327 (93.2)	162 (91.5)	103 (95.4)	82 (93.9)	
Yes	24 (6.8)	15 (9.5)	5 (4.8)	4 (6.1)	
**Calcium intake**					0.713
No	322 (91.7)	164 (92.7)	99 (91.7)	59 (89.4)	
Yes	29 (8.3)	13 (7.3)	9 (8.3)	7 (10.6)	
**Hypnotics**					0.048 *
No	329 (93.7)	166 (93.8)	105 (97.2)	58 (87.9)	
Yes	22 (6.3)	11 (6.2)	3 (2.8)	8 (12.1)	

Note: Middle-aged adults = 46–55 years; late middle-aged adults = 56–65 years; older adults = 66–75 years. Value are frequencies (percentages), unless otherwise specified. SD: standard deviation. Data was analyzed using Pearson correlation or ANOVA. * *p* < 0.05; ** *p* < 0.01.

**Table 3 healthcare-10-02250-t003:** Comparison of the prevalence of using different complementary and alternative medicines (CAMs) for relieving stress in study participants (*N* = 351).

Variable	*N* (%)				*p*-Value
	Total Population 351 (100)	Middle-Aged Adults 177 (50.4)	Late Middle-Aged Adults 108 (30.8)	Older Adults 66 (18.8)	
**CAM use**					0.045 *
**No**	116 (33.0)	48 (27.1)	40 (37.0)	28 (42.4)	
**Yes**	235 (67.0)	129 (72.9)	68 (63.0)	38 (57.6)	
**Number of CAMs used, mean (SD)**	3.5 ± 4.3	4.2 ± 4.7	3.1 ± 3.9	2.1 ± 3.0	<0.001 **
**Nutritional Approaches**					
Probiotics					0.193
No	283 (80.6)	136 (78.8)	91 (84.3)	56 (84.8)	
Yes	68 (19.4)	41 (23.2)	17 (15.7)	10 (15.2)	
Dietary supplements					
No	272 (77.5)	132 (74.6)	87 (80.6)	53 (80.3)	0.418
Yes	79 (22.5)	45 (25.4)	21 (19.4)	13 (19.7)	
Special diets					0.537
No	317 (90.3)	158 (89.3)	97 (89.8)	82 (93.9)	
Yes	34 (9.7)	19 (10.7)	11 (10.2)	4 (6.1)	
**Psychological Approaches**					
Mediation					0.010 *
No	276 (78.6)	133 (75.1)	82 (75.9)	81 (92.4)	
Yes	75 (21.4)	44 (24.9)	28 (24.1)	5 (7.8)	
Music therapies					<0.001 **
No	219 (62.4)	92 (52.0)	77 (71.3)	50 (75.8)	
Yes	132 (37.6)	85 (48.0)	31 (28.7)	16 (24.2)	
Relaxing therapies					0.271
No	266 (75.8)	130 (73.4)	81 (75.0)	55 (83.3)	
Yes	85 (24.2)	47 (26.6)	27 (25.0)	11 (16.7)	
**Physical Approaches**					
Acupuncture					0.838
No	305 (88.9)	154 (87.0)	95 (88.0)	56 (84.8)	
Yes	46 (13.1)	23 (13.0)	13 (12.0)	10 (15.2)	
Massage					0.008 **
No	242 (68.9)	109 (61.6)	80 (74.1)	53 (80.3)	
Yes	109 (31.1)	68 (38.4)	28 (25.9)	13 (19.7)	
Spinal manipulation					0.108
No	283 (74.9)	125 (70.6)	83 (76.9)	55 (83.3)	
Yes	88 (25.1)	52 (29.4)	25 (23.1)	11 (16.7)	
**Combinations of Psychological and Physical or Psychological and Nutritional Approaches**					
Yoga					<0.001 **
No	306 (87.2)	143 (80.8)	99 (91.7)	64 (97.0)	
Yes	45 (12.8)	34 (19.2)	9 (8.3)	2 (3.0)	
Tai-chi					0.93
No	327 (93.4)	166 (93.8)	100 (93.5)	61 (92.4)	
Yes	23 (6.8)	11 (6.2)	7 (6.5)	5 (7.6)	
Qigong					0.046 *
No	329 (94.0)	162 (91.5)	101	66 (100.0)	
Yes	21 (6.0)	15 (8.5)	6 (5.6)	0 (0.0)	
Art therapies					0.004 **
No	287 (81.8)	133 (75.1)	94 (87.0)	60 (90.9)	
Yes	64 (18.2)	44 (24.9)	14 (13.0)	6 (9.1)	
Aromatherapy					
No	294 (83.8)	136 (76.8)	96 (88.9)	62 (93.9)	<0.001 **
Yes	57 (16.2)	41 (23.2)	12 (11.1)	4 (6.1)	
**Other Complementary Health** **Approaches**					
Traditional Chinese medicine					0.997
No	292 (83.2)	147 (83.1)	90 (83.3)	90 (93.3)	
Yes	59 (16.8)	30 (16.9)	18 (16.7)	18 (6.7)	
Naturopathy					0.011 *
No	287 (81.8)	135 (78.3)	91 (84.3)	81 (92.4)	
Yes	64 (18.2)	42 (23.7)	17 (15.7)	5 (7.6)	
**Other**					
Functional medicine					
No	329 (94.0)	162 (91.5)	101 (94.4)	66 (100.0)	0.046 *
Yes	21 (6.0)	15 (8.5)	6 (5.6)	0 (0.0)	
Far infrared rays					0.014 *
No	315 (89.7)	152 (85.9)	98 (90.7)	65 (98.5)	
Yes	36 (10.3)	25 (14.1)	10 (9.3)	1 (1.5)	
Reading scriptures/*The Bible*					0.040 *
No	267 (76.1)	135 (76.3)	75 (69.4)	57 (86.4)	
Yes	84 (23.9)	42 (23.7)	33 (30.6)	9 (13.6)	
Fortune-telling					0.014 *
No	321 (91.7)	156 (88.1)	101 (94.4)	64 (97.0)	
Yes	29 (8.3)	21 (11.9)	8 (5.8)	2 (3.0)	

Note: Middle-aged adults = 46–55 years; late middle-aged adults = 56–65 years; older adults = 66–75 years. Value are frequencies (percentages), unless otherwise specified. SD: standard deviation. Data was analyzed using Pearson correlation or ANOVA. * *p* < 0.05; ** *p* < 0.01.

**Table 4 healthcare-10-02250-t004:** The log-binomial regression analyses of demographic and clinical factors influencing the use of complementary and alternative medicines (CAMs) in study participants (*N* = 351).

Variable	*N* (%)		Adjusted Prevalence Ratios (PR) (95% Confidence Interval)	*p*-Value
	NO Use	Use		
**CAM use**	**116 (33.0)**	**235 (67.0)**		
**Age**				
Middle-aged adults	48 (27.1)	129 (72.9)	1	
Late middle-aged adults	40 (37.0)	68 (63.0)	0.85 (0.72–1.00)	0.049 *
Older adults	28 (42.4)	38 (57.6)	0.86 (0.68–1.09)	0.85
**Religion**				
No	36 (40.4)	53 (59.6)	1	
Buddhism	43 (29.1)	105 (70.9)	1.18 (0.97–1.44)	0.094
Taoism	32 (36.0)	57 (64.0)	1.14 (0.92–1.43)	0.245
Christian/Other	5 (20.0)	20 (80.0)	1.43 (1.11–1.84)	0.006 **
**Vegetarian diets**				
No	39 (44.8)	48 (55.2)	1	
Irregularly	68 (28.5)	171 (71.5)	1.24 (1.01–1.52)	0.038 *
Regularly	9 (36.0)	16 (64.0)	1.09 (0.76–1.56)	0.657
**Prescription medication use**				
No	58 (40.8)	84 (59.2)	1	
Yes	58 (27.8)	151 (72.2)	1.26 (1.08–1.47)	0.003 **
**Hyperlipidemia**				
No	97 (30.6)	220 (69.4)	1	
Yes	19 (55.9)	15 (44.1)	0.62 (0.42–0.90)	0.012 *
**Allergic rhinitis**				
No	112 (34.1)	216 (65.9)	1	
Yes	4 (17.4)	19 (82.6)	1.27 (1.03–1.56)	0.027 *
**Music therapies**	**116 (33.0)**	**235 (67.0)**		
**Age**				
Middle-aged adults	92 (52.0)	85 (48.0)	1	
Late middle-aged adults	77 (71.3)	31 (28.7)	0.58 (0.42–0.80)	0.001 **
Older adults	50 (75.8)	16 (24.2)	0.58 (0.37–0.92)	0.019 *
**Religion**				
No	66 (74.2)	23 (25.8)	1	
Buddhism	80 (54.1)	68 (45.9)	1.86 (1.27–2.72)	0.001 **
Taoism	60 (67.2)	29 (32.6)	1.53 (0.98–2.38)	0.061
Christian/Other	13 (52.0)	12 (48.0)	2.16 (1.28–3.66)	0.004 **
**Vegetarian diets**				
No	66 (75.9)	21 (24.1)	1	
Irregularly	138 (57.7)	101 (42.3)	1.56 (1.05–2.30)	0.027 *
Regularly	15 (60.0)	10 (40.0)	1.39 (0.76–2.54)	0.286
**Hyperlipidemia**				
No	190 (59.9)	127 (40.1)	1	
Yes	29 (85.3)	5 (14.7)	0.42 (0.19–0.95)	0.036 *
**Massage**	**109 (31.1)**	**242 (68.9)**		
**Age**				
Middle-aged adults	109 (61.6)	68 (38.4)	1	
Late middle-aged adults	80 (74.1)	28 (25.9)	0.62 (0.44–0.89)	0.010 **
Older adults	53 (80.3)	13 (19.7)	0.58 (0.35–0.97)	0.036 *
**Religion**				
No	72 (80.9)	17 (19.1)	1	
Buddhism	94 (63.5)	54 (36.5)	2.02 (1.27–3.21)	0.003 **
Taoism	60 (67.4)	29 (32.6)	2.23 (1.34–3.71)	0.002 **
Christian/Other	16 (64.0)	9 (36.0)	2.05 (1.05–4.03)	0.037 *
**Smoking**				
No	199 (71.1)	81 (28.9)	1	
Irregularly	20 (54.1)	17 (45.9)	1.67 (1.09–2.53)	0.019 *
Regularly	23 (67.6)	11 (32.4)	1.38 (0.83–2.29)	0.219
**Vegetarian diets**				
No	74 (85.1)	13 (14.9)	1	
Irregularly	153 (64.0)	86 (36.0)	2.31 (1.36–3.92)	0.002 **
Regularly	15 (60.0)	10 (40.0)	2.67 (1.30–5.43)	0.007 **
**Spinal manipulation**	**88 (25.1)**	**283 (74.9)**		
**Occupation**				
No/Retired	87 (80.6)	21 (19.4)	1	
Teacher/Public employee	52 (75.4)	17 (24.6)	1.27 (0.72–2.24)	0.406
Salesmen/Attendance	29 (69.0)	13 (31.0)	1.66 (0.93–2.98)	0.089
Skilled/Professional	65 (76.5)	20 (23.5)	1.24 (0.71–2.15)	0.448
Other	30 (63.8)	17 (36.2)	1.79 (1.06–3.03)	0.029 *
**Alcohol use**				
No	193 (76.9)	58 (23.1)	1	
Irregularly	66 (71.7)	26 (28.3)	1.19 (0.80–1.77)	0.404
Regularly	4 (50.0)	4 (50.0)	1.98 (1.02–3.84)	0.045 *
**Perceived health status**				
**Fair**	21 (61.8)	13 (38.2)	1	
**Poor/Very poor**	144 (75.4)	47 (24.6)	0.66 (0.41–1.06)	0.087
**Good/Excellent**	98 (77.8)	28 (22.2)	0.59 (0.35–0.99)	0.044 *
**Relaxing therapies**	**85 (24.2)**	**266 (75.8)**		
**Religion**				
No	80 (89.9)	9 (10.1)	1	
Buddhism	102 (68.9)	46 (31.1)	2.90 (1.50–5.62)	0.002 **
Taoism	62 (69.7)	27 (30.3)	2.86 (1.43–5.73)	0.003 **
Christian/Other	22 (88.0)	3 (12.0)	1.18 (0.35–3.95)	0.794
**Reading scriptures/*The Bible***	**84 (23.9)**	**267 (76.1)**		
**Religion**				
No	85 (95.5)	4 (4.5)	1	
Buddhism	89 (60.1)	59 (13.9)	7.62 (2.84–20.48)	<0.001 **
Taoism	78 (87.6)	11 (12.4)	2.87 (0.96–8.59)	0.06
Christian/Other	15 (60.0)	10 (40.0)	8.47 (2.90–24.7)	<0.001 **
**Vegetarian diets**				
No	74 (85.1)	13 (14.9)	1	
Irregularly	182 (76.2)	57 (23.8)	1.48 (0.88–2.48)	0.136
Regularly	11 (44.0)	14 (56.0)	2.35 (1.23–4.49)	0.010 *
**Perceived health status**				
**Fair**	18 (52.9)	16 (47.1)	1	
**Poor/Very poor**	150 (78.5)	41 (21.5)	0.55 (0.36–0.84)	0.005 **
**Good/Excellent**	99 (78.6)	27 (21.4)	0.52 (0.33–0.82)	0.005 **

Other variables evaluated during the model development included sex, body mass index, marital status, number of children, education level, exercise, betel nut chewing, drinking coffee, drinking tea, drinking functional beverages, drinking milk, history of chronic disease, hypertension, presbyopia, myopia, sleep disorders, upset stomach, diabetes, anti-hypertensive, painkiller, upset stomach relief, muscle relaxant, calcium intakes, and hypnotics. * *p* < 0.05; ** *p* < 0.01.

## Data Availability

The data used in this study are available from the corresponding author upon reasonable request.

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
