# Peer review of "Complementary and Alternative Medicines Used by Middle-Aged to Older Taiwanese Adults to Cope with Stress during the COVID-19 Pandemic: A Cross-Sectional Survey"

_healthcare, 2022, doi:10.3390/healthcare10112250_

Round 1

Reviewer 1 Report (Previous Reviewer 1)

The authors have suscessfully addressed all my comments and suggestions.

This manuscript is a resubmission of an earlier submission. The following is a list of the peer review reports and author responses from that submission.

Round 1

Reviewer 1 Report

This review is interesting and important for understanding Complementary and Alternative Medicines Used to Cope with Stress during the COVID-19 Pandemic Period in Taiwanese Middle-aged to Older Adults. But several points must be improved.

Method Section

(Comment 1) Why authors excluded under 46 years old group? I think this may lead to biased research results.

(Comment 2) I recommend authors to re-write Method Section. In particular, I recommend authors to describe in detail definition of CAM treatment items in the process of developing the questionnaire.

e.g.

Materials and Methods

e.g. https://www.mdpi.com/1660-4601/19/6/3337/htm

https://bmccomplementmedtherapies.biomedcentral.com/articles/10.1186/s12906-017-1942-6

2.1. Method for survey

2.1.1. Survey sampling

2.1.2. Development of the survey form

2.1.3. Survey distribution

2.1.4. Data collection and analysis

(Comment 3) I recommend authors to supplement questionnaire as a supplementary file. This will contribute the reader's understanding and future follow-up studies.

(Comment 4) NCCIH (National Center for Complementary and Integrative Health) classified complementary medicine as follows (https://www.nccih.nih.gov/health/complementary-alternative-or-integrative-health-whats-in-a-name);

(1) Nutritional (e.g., special diets, dietary supplements, herbs, probiotics, and microbial-based therapies).

(1) Psychological (e.g., meditation, hypnosis, music therapies, relaxation therapies).

(3) Physical (e.g., acupuncture, massage, spinal manipulation).

(4) Combinations such as psychological and physical (e.g., yoga, tai chi, dance therapies, some forms of art therapy) or psychological and nutritional (e.g., mindful eating).

I recommend authors to describe the characteristics of Taiwanese CAM interventions compared to NCCIH's CAM classification.

Dicussion Section

(Comment 5) I recommend authors to remove "line 244-263" and re-write this part. The Discussion Section should be explained why older people have lower CAM utilization in the COVID pandemic. For example, reasons such as not going out during the COVID-19 outbreak or not visiting a CAM institutions.

(Comment 6) It is difficult to distinguish whether these results can be interpreted in relation to the COVID-19 pandemic or reflect the Taiwanese population's preference for CAM treatments. I recommend authors to supplement this point in Discussion Section.

Back matter

(Comment 7) I recommend authors to supplement IRB approval number and date.

e.g. https://www.mdpi.com/2072-6651/13/7/436/htm

Reviewer 2 Report

The manuscript investigated the factors influencing the use of complementary and alternative medicines (CAMS) to manage stress during the COVID-19 pandemic in Taiwan. However, the design and analysis may not be satisfying, thus making the conclusion not universal and not convincing enough. Some of the problems have been listed in the discussion part by the author, which are all serious issues need to be exactly solved rather than just put forward and compromised. I suggest major revisions be done before it is finally accepted.

1. The authors concluded that Population-specific mental health interventions using music can be developed to improve stress management outcomes during public health emergencies. Recent studies have revealed the close relationship between music listening and stress. For example, PMID: 34280999 could be cited.

2. Different types of music have different effects on stress. The influence of calm music, happy music and sad music on stress should be added to the manuscript or discussed.

3. The results came from 66 older adults, which is not enough. Please provide statistically analysis that how many samples are suitable to draw such a conclusion.

4. This manuscript has done a lot of research on factors influencing the use of complementary and alternative medicines (CAMS), but they are not reflected in the conclusion. The authors need to consult the relevant literature and discuss more. For example, PMID: 32792254 could be cited as a reference.

5. The CAPTION should include factors influencing the use of complementary and alternative medicines (CAMS).

6. There may be different types of staging systems for different cases. The authors should classify patients into high or low risk groups to make a score. PMID: 32690444 could be cited as a reference.

7. Furthermore, the authors only listed the limitations and the solutions and discussion should be stated. Beside, suggestions according to the result should also be further discussed in detail. 

Round 2

Reviewer 1 Report

The authors responded to comments and suggestions from reviewers. But several points must be improved.

(Comment 1) Autors supplemented survey sampling section. However, detail sampling process must be presented in Method Section. (line 100-106)

(1) How many people were surveyed and how many people responded?

(2) A sample of 351 people is considered small to represent Taiwanese I recommend authors to add why the author chose this number of respondents.

(Comment 2) Regarding reply's of comment 4, the characteristics of Taiwan CAM treatment should be presented in the Introduction or Discussion Section. This would justify the purpose of conducting this study*.

* Complementary and Alternative Medicines Used to Cope with Stress during the COVID-19 Pandemic Period in Taiwanese Middle-aged to Older Adults: A Cross-sectional Survey

(Comment 3) Regarding reply's of comment 5, author's responses and corrections are insufficient. I recommend authors to remove "line 270-290" and re-write this part.

- previous comment: I recommend authors to remove "line 244-263" and re-write this part. The Discussion Section should be explained why older people have lower CAM utilization in the COVID pandemic. For example, reasons such as not going out during the COVID-19 outbreak or not visiting a CAM institutions.

Reviewer 2 Report

I've checked the revised version and the authors have addressed my concerns perfectly. I have no other questions.
